# Daily Nutritional Supplementation with Vitamin D_3_ and Phenylbutyrate to Treatment-Naïve HIV Patients Tested in a Randomized Placebo-Controlled Trial

**DOI:** 10.3390/nu11010133

**Published:** 2019-01-10

**Authors:** Senait Ashenafi, Wondwossen Amogne, Endale Kassa, Nebiat Gebreselassie, Amsalu Bekele, Getachew Aseffa, Meron Getachew, Abraham Aseffa, Alemayehu Worku, Ulf Hammar, Peter Bergman, Getachew Aderaye, Jan Andersson, Susanna Brighenti

**Affiliations:** 1Center for Infectious Medicine (CIM), Department of Medicine Huddinge, Karolinska Institutet, Alfred Nobels Allé 8 (ANA8), 141 52 Huddinge, Sweden; Senait.Ashenafi.Betemariam@ki.se (S.A.); nebiatye@yahoo.com (N.G.); jan.p.andersson@sll.se (J.A.); 2Department of Internal Medicine, School of Medicine, College of Health Sciences, Tikur Anbessa University Hospital and Addis Ababa University, Addis Ababa, Ethiopia; wonamogne@yahoo.com (W.A.); endalekl@yahoo.com (E.K.); amsalubekele2016@gmail.com (A.B.); merget2004@yahoo.com (M.G.); getadera@yahoo.com (G.A.); 3Armauer Hansen Research Institute (AHRI), Addis Ababa, Ethiopia; aseffaa@gmail.com; 4Department of Radiology, School of Medicine, College of Health Sciences, Tikur Anbessa University Hospital and Addis Ababa University, Addis Ababa, Ethiopia; g_asefa@yahoo.com; 5Department of Public Health, School of Medicine, College of Health Sciences, Tikur Anbessa University Hospital and Addis Ababa University, Addis Ababa, Ethiopia; alemayehuwy@yahoo.com; 6Institute of Environmental Medicine (IMM), Karolinska Institutet, 171 77 Stockholm, Sweden; ulf.hammar@ki.se; 7Clinical Microbiology, Department of Laboratory Medicine (Labmed), Karolinska Institutet, Alfred Nobels Allé 8 (ANA8), 141 52 Huddinge, Sweden; peter.bergman@ki.se; 8Department of Medicine Huddinge, Division of Infectious Diseases, Karolinska University Hospital Huddinge, 141 86 Stockholm, Sweden

**Keywords:** vitamin D, nutrition, supplementation, HIV infection, clinical trial, immunity

## Abstract

Poor nutritional status is common among human immunodeficiency virus (HIV)-infected patients including vitamin D (vitD_3_) deficiency. We conducted a double-blinded, randomized, and placebo-controlled trial in Addis Ababa, Ethiopia, to investigate if daily nutritional supplementation with vitD_3_ (5000 IU) and phenylbutyrate (PBA, 2 × 500 mg) could mediate beneficial effects in treatment-naïve HIV patients. Primary endpoint: the change in plasma HIV-1 comparing week 0 to 16 using modified intention-to-treat (mITT, *n* = 197) and per-protocol (*n* = 173) analyses. Secondary endpoints: longitudinal HIV viral load, T cell counts, body mass index (BMI), middle-upper-arm circumference (MUAC), and 25(OH)D_3_ levels in plasma. Baseline characteristics were detectable viral loads (median 7897 copies/mL), low CD4^+^ (median 410 cells/µL), and elevated CD8^+^ (median 930 cells/µL) T cell counts. Most subjects were vitD_3_ deficient at enrolment, but a gradual and significant improvement of vitD_3_ status was demonstrated in the vitD_3_ + PBA group compared with placebo (*p* < 0.0001) from week 0 to 16 (median 37.5 versus 115.5 nmol/L). No significant changes in HIV viral load, CD4^+^ or CD8^+^ T cell counts, BMI or MUAC could be detected. Clinical adverse events were similar in both groups. Daily vitD_3_ + PBA for 16 weeks was well-tolerated and effectively improved vitD_3_ status but did not reduce viral load, restore peripheral T cell counts or improve BMI or MUAC in HIV patients with slow progressive disease. Clinicaltrials.gov NCT01702974.

## 1. Introduction

Human immunodeficiency virus (HIV)/acquired immunodeficiency syndrome (AIDS) is a key challenge for global health and most of the world’s population living with HIV infection is in Africa. HIV covers multiple stages, from initial and acute infection to chronic infection and further progression to advanced HIV disease or full-blown AIDS. In addition to the different phases of HIV infection, the rate of disease progression may be substantially different among HIV-infected individuals [1,2]. HIV-infected individuals can be categorized as progressors or controllers, and several subgroups within these disease phenotypes exist [3]. While plasma HIV viral load predicts disease progression, CD4^+^ T cell counts in peripheral blood are useful to determine immune status and stage of HIV infection [4]. Some HIV patients are called long-term slow progressors, as they maintain normal CD4^+^ T cell counts and low but detectable HIV viral loads, usually below 10,000 copies/mL [5]. In contrast, the majority of untreated HIV-infected individuals experience intermediate disease progression characterized by a decline in CD4^+^ T cells counts and a concomitant increase in HIV viral load over time [1,2]. It is well-established that HIV is associated with a massive depletion of CD4^+^ T helper cells, particularly in the gut [6], but without antiretroviral therapy (ART), CD4^+^ T cell counts also decline in the peripheral blood. CD4^+^ levels below 200 cells/µL is associated with immunodeficiency, resulting in an increased susceptibility to a wide variety of opportunistic infections [7]. World Health Organization (WHO) guidelines from 2015 recommend initiation of ART to HIV-infected individuals as soon as possible, regardless of CD4^+^ T cell counts, to reduce the morbidity and mortality associated with HIV infection [8]. While the previous guidelines instructed to start ART at CD4^+^ T cell counts < 350 cells/µL [3], these recommendations were revised based on evidence from clinical trials showing that earlier initiation of ART could delay a decline in CD4^+^ T cell counts and prevent immunological deterioration [9,10].

Maintaining stable CD4^+^ T cell counts and low viral loads will reduce the risk of HIV complications and increase patient’s quality of life as well as life expectancy. The immune system is under great stress and a balanced diet including a variety of nutrients can strengthen immunity and maintain body weight [11]. Here, nutritional supplements such as vitamin D_3_ (vitD_3_) and phenylbutyrate (PBA), possess pleiotropic immunomodulatory functions that could improve innate mucosal immunity and simultaneously prevent chronic immune activation and dysregulation caused by adaptive immunity [12]. VitD_3_ is a hormone that can be produced in the skin after exposure of UVB-light or obtained via the diet [13]. PBA is an aromatic short-chain fatty acid and a well-known histone deacetylase (HDAC) inhibitor [14]. Together, vitD_3_ and PBA can induce expression of the human cathelicidin, LL-37, which is an antimicrobial peptide with broad activity [15], including potential anti-viral properties [16]. VitD_3_ and PBA can also enhance autophagy, which is a physiological process known to enhance destruction of intracellular bacteria [17] and viruses [18]. HIV can inhibit autophagy, which prevents lysosomal degradation of HIV proteins within autophagosomes and also modulates the function of different immune cells [19]. Importantly, LL-37 is essential for vitD_3_-induced autophagic flux and inhibition of HIV replication in human macrophages in vitro [20,21]. Thus, enhanced viral destruction may simultaneously reduce microbial translocation and inflammation at mucosal sites.

Given these previous data, we hypothesized that daily nutritional supplementation using vitD_3_ + PBA could reduce viral replication and restore immune and nutritional status in HIV infection. To test this, we performed a double-blind, randomized and placebo-controlled trial in Ethiopia. At the time this clinical trial was conducted, the national guidelines in Ethiopia were to initiate ART in HIV patients with clinical symptoms and a CD4^+^ T cell count <350 cells/µL. Therefore, ART-naïve HIV-positive individuals with slow progressive disease were enrolled and the response to treatment was evaluated using HIV viral load, assessed at baseline, and compared to 16 weeks of treatment. Secondary endpoints included longitudinal analyses (week 0, 4, 8, 16, and 24) of HIV viral load, CD4^+^ and CD8^+^ T cell counts, body mass index (BMI), middle-upper-arm circumference (MUAC), and 25-hydroxy-vitD_3_ (25(OH)D_3_) levels in plasma. Clinical symptoms were recorded at follow up to monitor adverse events (AEs).

## 2. Materials and Methods

For details on the methods, please see the online Appendix A.

### 2.1. Study Design

This study was a randomized, double-blind, placebo-controlled, clinical trial conducted at the pre-ART clinic, Department of Internal Medicine, School of Medicine, College of Health Sciences, Tikur Anbessa University Hospital in Addis Ababa, Ethiopia after ethical approval in Ethiopia and Sweden. The study was registered at www.clinicaltrials.gov, NCT01702974, prior to inclusion of the first patient.

### 2.2. Patients

Inclusion criteria: Adult HIV-positive patients >18 years not subjected to ART with CD4^+^ T cells counts >350 cells/µL and detectable plasma viral loads >1000 copies/mL. Exclusion criteria: Patients on ART or other antimicrobial drugs (including trimethoprim-sulfamethoxazole), antimicrobial drug treatment in the past month, hypercalcaemia (serum calcium >3.0 mmol/L), pregnancy and breast-feeding, liver or renal diseases, malignancies, or treatment with cardiac glycosides. For comparison of baseline parameters, clinical and laboratory assessments were also performed on *n* = 52 HIV-negative controls who did not take part in the clinical trial. All patients and controls provided written and signed informed consent before enrolment.

### 2.3. Interventions

This was a two-arm intervention trial using daily adjunct therapy with vitD_3_ and PBA over 16 weeks. Patients were randomized to receive daily oral supplementation using the following dosing scheme: (1) 5000 IU vitD_3_ (five tablets once daily) and 500 mg PBA (one tablet twice daily), or (2) vitD_3_ placebo and PBA placebo tablets. Good manufacturing practice-produced vitD_3_ tablets (Vigantoletten) and matching placebo were donated by Merck KGaA (Darmstadt, Germany); PBA (Sodium Phenylbutyrate) and matching placebo were obtained from Scandinavian Formulas Inc. (Sellersville, PA, USA).

### 2.4. Randomization and Masking

Subjects were randomized in a one-to-one allocation ratio using computer-generated randomization codes and block randomization with a block size of ten (Karolinska Trial Alliance, Stockholm, Sweden), to ensure that in each block, five subjects were randomized to vitD_3_ + PBA and the other five subjects to placebo. Pharmacists at the Tikur Anbessa Hospital prepared the study medication and provided the randomization codes that assigned the patients to vitD_3_ + PBA or placebo treatment. Patients were recruited by senior consultants and a health officer, and they were all blinded to the randomization.

### 2.5. Outcome Measures

The primary endpoint was the change in HIV viral load in plasma assessed at 16 weeks compared with baseline (time point 0). Secondary endpoints included longitudinal assessments of HIV viral load, CD4^+^ and CD8^+^ T cells counts, BMI and MUAC (week 0, 4, 8, 16, and 24) and 25(OH)D_3_ levels in plasma (week 0, 4, 8 and 16).

### 2.6. Procedures

Blood samples were collected for the described laboratory analyses that were conducted at the International Clinical Laboratory (ICL), in Addis Ababa, Ethiopia. HIV testing was performed according to the national guidelines, while HIV-1 RNA levels in plasma were quantified using the Abbott RealTime HIV-1 Viral Load assay (Abbott Laboratories, Chicago, IL, USA). The CD4^+^/CD8^+^ T cell counts was determined using BD FACSCount (BD Biosciences, San Jose, NJ, USA). Levels of 25(OH)D_3_ in plasma were analyzed at the Department of Clinical Chemistry, Karolinska University Hospital in Stockholm, Sweden using a chemiluminescence immunoassay (CLIA) on a LIAISON-instrument (DiaSorin Inc., Stillwater, MN, USA), detectable range 7.5–175 nmol/L, CV 2–5%. Safety assessments to monitor AEs included clinical examinations (week 4, 8, 16, and 24) and blood chemistry analysis (week 0, 4, 8, and 16) to measure liver and kidney function, and calcium/phosphate homeostasis.

### 2.7. Statistical Analysis

Based on previous publications, we expected a spread in the average HIV viral load between 1000 and 100,000 copies/mL. We hypothesized that treatment with vitD_3_ + PBA would reduce viral load with 25% (corresponding to a log reduction of approximately 0.125), with no change in the control group. Based on a previous study, we estimated that the standard deviation of the longitudinal change in logarithmic viral load (log reduction) would be approximately 0.276 in the control group [22]. To account for variations in treatment response, a larger standard deviation of 0.32 was anticipated in the vitD_3_ + PBA treated group. Based on this assumption, a sample size of approximately 90 patients per group was required to detect the desired effect (80% power, alpha = 0.05, two-sided test). With a calculated dropout rate of 15%, 103 patients/arm = 206 patients in total were enrolled. Results were analyzed following the intention-to-treat (ITT)-concept, using multiple imputation by chained equations to impute outcomes for persons lost to follow-up. In addition, per-protocol analyses, which included all HIV patients who completed the intervention, was used. Primary and secondary analyses, as well as post hoc analyses, were conducted using linear regression. Both crude and adjusted analyses were made. The covariates adjusted for were age (years), gender (male/female), CD4^+^ T cell counts, and baseline value of the outcome, i.e., HIV viral load (log_10_ copies/mL). Those variables were selected a priori to increase the precision of our estimates, since we believed them to be associated with the outcome [23]. A *p*-value <0.05 was considered significant. Analyses were conducted using IBM SPSS Statistics 20.0 and Stata 13 (StataCorp, College Station, TX, USA).

## 3. Results

### 3.1. Enrolment

Initially, 562 HIV-infected individuals were screened for eligibility from January 2013 to May 2015 as described in the CONSORT chart (Figure 1). After randomization of 278 patients, laboratory testing including primarily HIV viral load in plasma, confirmed that 81 enrolled patients did not fulfill the pre-defined exclusion criteria (low HIV viral load <1000 copies/mL, *n* = 78; low CD4^+^ T cell counts <200 cells/mL, *n* = 2 and chronic liver disease, *n* = 1). The remaining 197 subjects constituted the modified ITT (mITT) cohort, allocated to vitD_3_ + PBA (*n* = 95) or placebo (*n* = 102) treatment. A total of 24 patients discontinued intervention or were lost to follow-up (dropout rate = 12.2%). Thus, 173 patients completed the treatment per-protocol, allocated to vitD_3_ + PBA (*n* = 85) or placebo (*n* = 88) treatment. 

### 3.2. Baseline Characteristics

Baseline data are presented in Table 1 and Table 2, and in Appendix A. The majority (80%) of enrolled study subjects were females with a median HIV viral load in plasma of 7897 copies/mL (Table 1). While more than half of the HIV patients had a viral load <10,000 copies/mL, only 19 (9.6%) patients had a viral load above 100,000 copies/mL (Table 1). Comparing HIV-positive study subjects with a group of HIV-negative controls, confirmed that most of the baseline variables were abnormal in the HIV patients, regardless if the analyses were adjusted for gender or not (Table 1). Patients had low CD4^+^ (median 410 cells/µL) and elevated CD8^+^ (median 930 cells/µL) T cell counts in comparison with HIV-negative controls who had significantly higher CD4^+^ (median 593 cells/µL, *p* < 0.001) but lower CD8^+^ (median 500 cells/µL, *p* < 0.001) T cell counts (Table 1). Accordingly, the CD4/CD8 ratio was significantly reduced in HIV-positive individuals (Table 1). At baseline, there was a significant inverse correlation (*r* = −0.17, *p* = 0.018) between HIV viral load and CD4^+^ T cell counts in HIV-positive patients (Appendix A). The weight of enrolled subjects was significantly lower compared with HIV-negative individuals (*p* < 0.001) and consistently also BMI (*p* = 0.26) and MUAC (*p* = 0.02) were relatively lower in HIV-positive patients (Table 1). These results confirmed that enrolled study participants had progressive HIV infection, but with viral loads in the lower range.

Among enrolled HIV-positive study subjects, baseline variables were not statistically different between the placebo and vitD_3_ + PBA group (Table 2). Plasma 25(OH)D_3_ concentrations were low, around 38 nmol/L and thus most HIV-positive patients were vitD_3_ deficient (69.5%) [13] or insufficient (21.3%) (Table 1 and Table 2).

### 3.3. Primary Endpoint: HIV Viral Load

Longitudinal assessments of HIV viral load are demonstrated in Figure 2 and the differences and 95% CI are shown in Table 3. Overall, there was no difference in HIV viral load comparing vitD_3_ + PBA treatment to placebo, and the viral load was maintained at similar levels in both groups during the study period of 24 weeks. In the adjusted per-protocol analysis, HIV viral load was significantly reduced at week 24 in the placebo group (*p* = 0.031). However, the change in absolute viral copies was small (0.35 log_10_) including a broad confidence interval, indicating that this effect was not clinically relevant; neither could sub-group analyses of HIV patients with 25(OH)D_3_ ≤ 50 nmol/L or HIV viral load >5000 copies/mL (data not shown) show a clinically significant difference in HIV viral load comparing treatment with placebo.

### 3.4. Secondary Endpoints: CD4^+^ and CD8^+^ T Cell Counts, BMI, MUAC, and vitD_3_ Status

Longitudinal analysis showed no significant effect of vitD_3_ + PBA treatment on either CD4^+^ or CD8^+^ T cell counts (Figure 3a,b). Neither were any relevant changes in BMI or MUAC detected (Figure 3c,d).

Most subjects had low plasma vitD_3_ levels at baseline (Table 2) that increased significantly (*p* < 0.0001) in the vitD_3_ + PBA group compared with placebo at week 4 (mean 82.7 vs 41.8 nmol/L), week 8 (mean 103.4 vs 40.3 nmol/L), and week 16 (mean 120.4 vs 43.7 nmol/L) (Figure 4). Significantly improved vitD_3_ status in the vitD_3_ + PBA group indicated response to study treatment and good adherence.

### 3.5. Adverse Events

Clinical and laboratory AEs were monitored using clinical examination and blood chemistry analyses. The major clinical AEs observed at follow-up (week 4–16) were mostly mild and are listed in Table 4. Overall, AEs were reported in 39 (38.2%) placebo and 35 (36.8%) vitD_3_ + PBA treated patients. A total of 53 and 48 AEs were detected in the placebo and treatment group, respectively, meaning some patients experienced several AEs during follow up. There were no major differences in the types, manifestation or numbers of AEs between placebo and vitD_3_ + PBA treatment. The most common symptoms in general were other infections than HIV (placebo, *n* = 21 versus vitD_3_ + PBA, *n* = 24). In the placebo group, patients commonly experienced urinary and respiratory tract infections, cough, diarrhea, abdominal cramps, and dyspepsia. In the treatment group, urinary tract infections dominated followed by dyspepsia. Observed over time, we did not detect any clinically relevant changes in blood chemistry (calcium, phosphate, albumin, or creatine) related to the intervention (Appendix A). No severe clinical or laboratory AEs were reported in the study cohort (data not shown).

## 4. Discussion

Nutritional supplementation could represent a simple and low-cost alternative to recover immune status in HIV-positive individuals, particularly in resource-limited settings. This study showed that daily supplementation with vitD_3_ + PBA for 16 weeks failed to improve HIV viral loads, CD4^+^ or CD8^+^ T cells counts, BMI or MUAC in ART-naïve HIV-positive patients, although the intervention group responded with elevated vitD_3_ levels within 4 weeks, rapidly correcting vitD_3_ deficiency. Baseline characteristics confirmed that enrolled patients presented typical hallmarks of HIV infection and slow progressive disease, including viral loads mostly in the lower range, low CD4^+^ T cell counts, elevated CD8^+^ T cell counts and an inverse correlation between HIV virus and CD4^+^ T cells. Moreover, HIV patients experienced significant weight loss in addition to low BMI and MUAC. Administration of vitD_3_ + PBA daily for 16 weeks was safe and well-tolerated and the numbers of clinical AEs were not different between the groups.

This study has several strengths. Most HIV-positive subjects were vitD_3_-deficient at baseline, providing a rationale for vitD_3_ supplementation. Numerous studies report that vitD_3_ deficiency is common among HIV-infected patients [24,25,26], although vitD_3_ deficiency is often observed in similar frequencies in the general population [27,28,29]. It has been shown that HIV viral load was significantly higher among HIV patients on ART with insufficient vitD_3_ levels [30]. Interestingly, low vitD_3_ status among healthy South-African adults during the winter season enhanced viral replication after in vitro exposure of blood cells to HIV-1 [31]. However, HIV replication in vitro was attenuated after high-dose oral weekly vitD_3_ supplementation in vivo that also increased circulating white blood cells and reversed winter-associated anemia [31]. Another study demonstrated that low vitD_3_ status in HIV-infected pregnant women in Tanzania was significantly associated with HIV disease progression, all-cause mortality, and severe anemia, while no change in CD4^+^ or CD8^+^ T cell counts was observed at follow-up [32]. Women with low vitD_3_ also had a lower BMI and enhanced risk of acute respiratory tract infections as well as thrush compared to women with adequate vitD_3_ levels [33]. In contrast, a recent study found no association between vitD_3_ metabolites in blood and CD4^+^ T cell recovery in HIV-positive males initiating ART [34]. Overall, dark skin, female sex, winter season, low CD4^+^ T cell counts and ongoing ART have been identified as risk factors for severe vitD_3_ deficiency [27,35]. Initiation of ART has been shown to contribute to elevated levels of vitD_3_ binding protein in plasma, which may reduce bioavailability of vitD_3_ in target cells and tissues [36]. Accordingly, vitD_3_ deficiency is linked to more inflammation and immune activation [24,37], low peripheral blood CD4^+^ T cells [38], faster progression of HIV disease, and shorter survival time in HIV-infected patients [39].

Another strength of this study was that daily doses of vitD_3_ was administered together with PBA instead of using a bolus regimen. High-dose bolus vitD_3_ supplementation causes large fluctuations in circulating 25(OH)D_3_ concentrations [40], which may lead to a dysregulation of 1α-hydroxylase activity that ultimately reduce the conversion to active 1,25(OH)_2_D_3_ available in immune cells and tissues [41]. Thus, although it is convenient to give patients large bolus doses of vitD_3_ to increase adherence, lower doses given more often (daily or weekly) may be required to maintain stable levels of biologically active vitD_3_ that could induce protective immunity [12]. Using this treatment protocol, we recently demonstrated that daily vitD_3_ + PBA can support standard chemotherapy and improve clinical symptoms in patients with active pulmonary tuberculosis (TB) [42].

Our study also has some limitations. First, there was a skewed gender distribution in the study cohort since the majority of enrolled subjects were women. However, the proportion of females was similar in the placebo and the vitD_3_ + PBA group and the adjusted analysis corrected for this imbalance. A retrospective analysis of HIV-positive adults enrolled at 56 different health facilities in Ethiopia 2006–2011, confirmed that the majority of the HIV population were females [8], which suggest that the gender distribution in our cohort is representative of people with HIV in the country. In addition, we found that the uninfected control group contained more males than females, and the controls had significantly lower vitD_3_ status compared with the HIV-infected subjects. Consistently, a recent study comparing vitD_3_ status in males and females, enrolled in a large Indian cohort (3879 participants), found that vitD_3_ status was significantly lower in males who also had a higher incidence of vitD_3_ deficiency (<30 nmol/L) [43]. The importance of gender in clinical vitD_3_ trials is rather unexplored but may affect baseline vitD_3_ status as well as the response to vitD_3_ supplementation and should therefore be further explored.

Another weakness was that we could not dissect the potential effects of vitD_3_ and the HDAC inhibitor PBA separately, as we designed a two-arm intervention trial to increase the power of the study. In this context, it has previously been discovered that histone acetylation may increase the accessibility of the chromatin that results in enhanced HIV RNA transcription [44]. Accordingly, treatment of CD4^+^ T cells from HIV patients with other HDAC inhibitors such as vorinostat [45] and romidepsin [46] reactivate latent HIV virus with the rationale that pharmacological activation of HIV in combination with suppressive ART may result in depletion of latently infected, resting CD4^+^ T cells. However, it is possible that in the absence of ART, the in vivo effects of PBA counterbalanced potential positive effects of vitD_3_. A four-arm intervention trial would have shed additional light on the individual versus combined effects of PBA and vitD_3_. Another limitation of our study was that the inclusion criteria of HIV subjects only allowed enrollment of ART-naïve patients with detectable HIV viral loads >1000 copies/mL and stable CD4^+^ T cell counts >350 cells/µL. While our results failed to demonstrate an effect of vitD_3_ + PBA treatment in this group of patients, it is possible that HIV-positive patients with advanced HIV including high viral loads <100.000 copies/mL and low CD4^+^ T counts <200 cells/µL would be more likely to benefit from adjunct vitD_3_ + PBA treatment in the presence of ART. Secondary analyses of inflammation and T cell activation could add additional information on the response to vitD_3_ + PBA treatment in these patients.

Previous trials using adjunct vitD_3_ supplementation in HIV have demonstrated reduction in HIV viral load [47], enhanced CD4^+^ T cell recovery [48], increased frequencies of antigen-specific T cells expressing macrophage inflammatory protein (MIP)-1β and plasma levels of LL-37 [49], and decreased Th17–Treg ratios [50]. It has been shown that vitD_3_ insufficiency may impair CD4^+^ T cell recovery after initiation of ART in HIV-infected women [51]. However, vitD_3_ bolus dosing bimonthly for a year [52], or weekly for 6 months [53] did not improve CD4^+^ T cell counts or HIV viral load among HIV-infected children and adolescents. Another trial showed that serum from HIV-positive patients treated with high daily doses of vitD_3_ for 52 weeks demonstrated elevated TLR2/1 ligand-induced expression of LL-37 mRNA in human monocytes [54]. This effect was not observed at earlier time-points and was independent of ART, suggesting that longer periods of vitD_3_ supplementation are required to improve antimicrobial immunity [54]. Apart from the effects on immune functions, vitD_3_ supplementation can also prevent ART-induced reduction of bone mineral density in adolescents [55] and adults [56], promote bone formation [57] and improve neuromuscular motor skills [58] in HIV-infected individuals, which may decrease the risk of osteoporosis and fractures. Therefore, the benefits for sufficient vitD_3_ levels in HIV patients may reach beyond the outcomes investigated in this trial.

## 5. Conclusions

Daily nutritional supplementation with vitD_3_ and PBA safely and rapidly corrected vitD_3_ deficiency in ART-naïve HIV-positive patients, but failed to demonstrate positive effects on HIV viral load, T cell counts, BMI, or MUAC. Future trials need to be conducted to explore the combination effects of these immunomodulatory compounds in the presence of ART.

## Figures and Tables

**Figure 1 nutrients-11-00133-f001:**
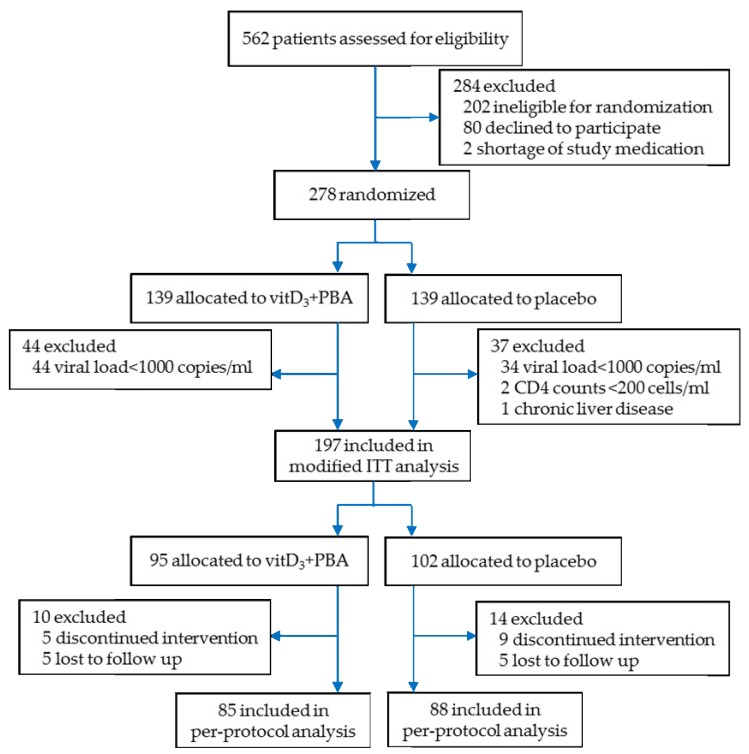
Trial profile. Flow diagram of patients screened for human immunodeficiency virus (HIV) infection. Patients ineligible for randomization included age <18 years (*n* = 27), pregnancy or breast feeding (*n* = 40), home parenteral nutrition (*n* = 56), low HIV viral loads <1000 copies/mL (*n* = 60), low CD4^+^ T cell counts <350 cells/µL (*n* = 16), initiated antiretroviral therapy (ART; *n* = 1), liver diseases (*n* = 2), shortage of study medication (*n* = 2) and patients who declined to participate (*n* = 80). Delayed laboratory results that were received after randomization confirmed that some patients had low HIV viral loads <1000 copies/mL (*n* = 78), liver disease (*n* = 1), and low CD4^+^ T cells counts <200 cells/µL (*n* = 2). Discontinued intervention included patients who initiated ART (*n* = 8), pregnancy (*n* = 3), tuberculosis infection (*n* = 2), and adverse events (*n* = 1). Patients who dropped out from the study included patients who withdrew their consent (=6), moved from the study area (*n* = 3), or could not be reached (*n* = 1).

**Figure 2 nutrients-11-00133-f002:**
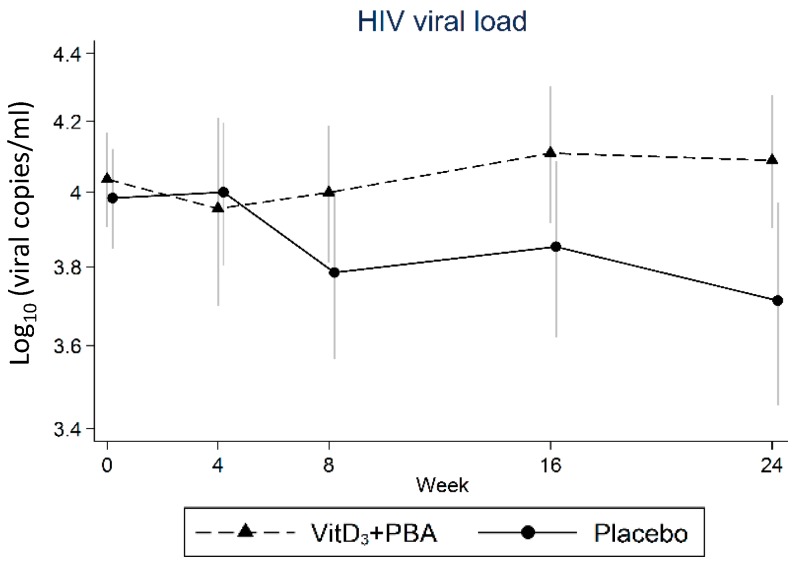
Primary efficacy analyses. HIV viral load was assessed at baseline and at weeks 4, 8, 16, and 24 after initiation of vitD_3_ + PBA supplementation. The efficacy analysis included comparison of log HIV viral load in the vitD_3_ + PBA and placebo treatment groups between week 0 and week 8. Crude data from the mITT cohort are presented as the mean and 95% CI. The solid line represents placebo while the dotted line represents vitD_3_ + PBA treatment.

**Figure 3 nutrients-11-00133-f003:**
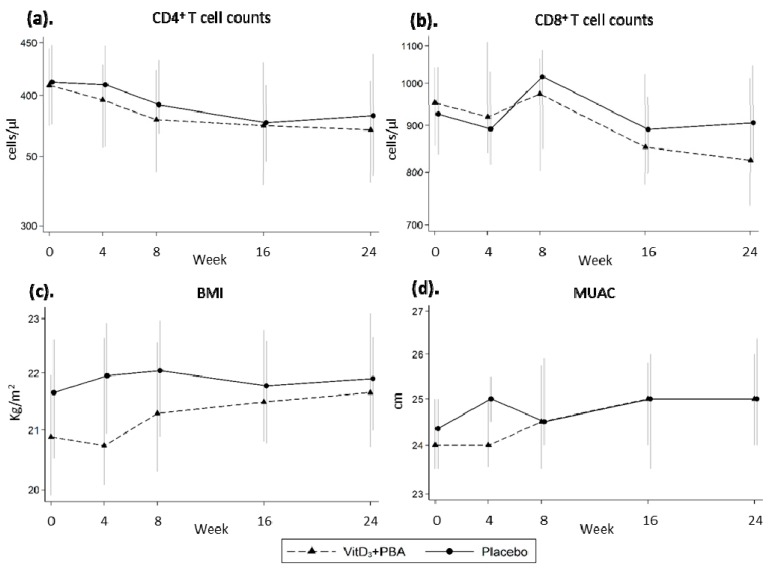
Secondary analyses. Peripheral CD4^+^ and CD8^+^ T cell counts, BMI, and MUAC were assessed at baseline and at weeks 4, 8, 16, and 24 after initiation of vitD_3_ + PBA supplementation. (**a**) CD4^+^ T cell counts, (**b**) CD8^+^ T cell counts, (**c**) BMI, (**d**) MUAC. Crude data from the mITT cohort are presented as the median and 95% CI. In **a**–**d**, the solid line represents placebo while the dotted line represents vitD_3_ + PBA treatment. BMI, Body Mass Index; MUAC, Mid-Upper-Arm Circumference.

**Figure 4 nutrients-11-00133-f004:**
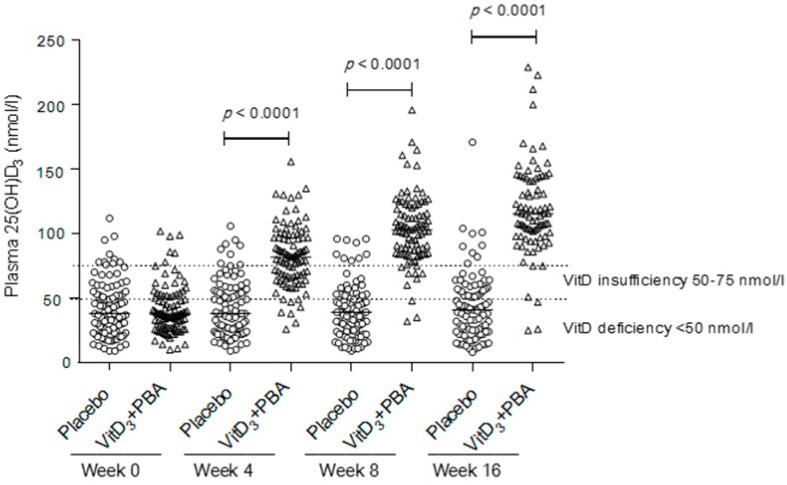
Vitamin D status. Plasma levels of 25(OH)D_3_ were assessed at baseline and at weeks 4, 8, and 16 after initiation of vitD_3_ + PBA supplementation. Placebo (circles) compared with the vitD_3_ + PBA group (triangles) is shown in a scatter dot plot. The solid line indicates the median, and the dashed lines mark the different thresholds for vitD_3_ deficiency and insufficiency [13].

**Table 1 nutrients-11-00133-t001:** Baseline characteristic in HIV-positive patients versus HIV-negative controls.

Variables ^1^ (mITT, *n* = 197)	HIV-pos Patients (*n* = 197)	HIV-neg Controls ^2^ (*n* = 52)	*p*-Value ^3^
Gender (M/F) (no/%)	40 (20)/157 (80)	33 (63)/19 (37)	<0.001
Age (years)	30 (26, 38)	36 (26, 46)	0.037
HIV viral load (copies/mL)	7897 (3116, 27,792)	-	-
HIV viral load < 10.000 (no/%)	102 (51.8)		
HIV viral load 10.000–100.000	76 (38.6)		
HIV viral load > 100.000	19 (9.6)		
CD4 T cell counts (cells/µL)	410 (324, 510)	593 (495, 767)	<0.001
CD8 T cell counts (cells/µL)	930 (691, 1253)	500 (350, 668)	<0.001
CD4/CD8 ratio	0.44	1.19	<0.001
Weight (kg)	54 (50, 64)	68 (59, 78)	<0.001
Weight loss (no/%)	58 (29%)	-	-
BMI (kg/m^2^)	21.1 (19.1, 23.9)	23.0 (19.5, 24.3)	0.260
MUAC (cm)	24.2 (23.0, 26.5)	25.8 (24.0, 28.0)	0.022
Pulse rate/min	78 (76, 82)	78 (73, 88)	0.250
Respiratory rate/min	18 (17, 19)	18 (16, 20)	0.770
25(OH)D_3_ nmol/L	38 (26, 52)	28 (19, 44.5)	0.003
Deficiency < 50 nmol/L (no/%)	137 (69.5)	45 (86.5)	0.029
Insufficiency 50–75 nmol/L	42 (21.3)	7 (13.5)	
Sufficiency > 75 nmol/L	16 (8.2)	0 (0)	

^1^ Data are *n* (%) or median (25th, 75th percentile). ^2^ Baseline data in HIV-negative controls, not included in the clinical trial. ^3^ Statistical significance between HIV-positive (*n* = 197) and HIV-negative (*n* = 52) individuals using Mann–Whitney U-test for continuous variables and chi-square tests for categorical variables. mITT, modified intention-to-treat; BMI, Body Mass Index; MUAC, Mid-Upper-Arm Circumference; 25(OH)D_3_, 25-hydroxyvitamin D.

**Table 2 nutrients-11-00133-t002:** Baseline characteristics in placebo versus vitD_3_ + PBA.

Variables (mITT, *n* = 197)	Placebo (*n* = 102)	VitD_3_ + PBA (*n* = 95)
Gender (M/F) (no/%)	23 (23)/79 (77)	17 (18)/78 (82)
Age (years)	30 (27, 38)	30 (25, 39)
HIV viral load (copies/mL)	7008 (2630, 23,267)	10,037 (3443, 32,445)
HIV viral load >100.000 (no/%)	11 (10.8)	8 (8.4)
HIV viral load < 10.000 (no/%)	57 (55.9)	45 (47.4)
CD4 T cell counts (cells/µL)	412 (340, 505)	409 (320, 517)
CD8 T cell counts (cells/µL)	927 (661, 1265)	952 (733, 1207)
CD4/CD8 ratio	0.44	0.43
Weight (kg)	56 (50, 66)	53 (50, 62)
Weight loss (no/%)	31 (30%)	27 (28%)
BMI (kg/m^2^)	21.6 (19, 24.6)	20.9 (19.1, 23.6)
MUAC (cm)	24.3 (23.0, 27.0)	24.0 (23.0, 26.0)
Pulse rate/min	78 (75, 82)	78 (76, 82)
Respiratory rate/min	17 (16, 18)	18 (17, 20)
WBC (SI units)	5.2 (4.1, 6.4)	5.5 (4.3, 6.7)
ESR (mm/h)	31 (18, 42)	34 (20, 48)
Hemoglobin (g/L)	14 (12, 15)	14 (12, 15)
Calcium (g/L)	9 (8.4, 9.5)	8.8 (8.3, 9.4)
Albumin (mg/dl)	4 (3.7, 4.2)	4 (3.6, 4.2)
25(OH)D_3_ nmol/L	38 (24, 53)	37.5 (27, 52)
Deficiency < 50 nmol/L (no/%)	69 (68.3)	69 (73.4)
Insufficiency 50–75 nmol/L	22 (21.8)	19 (20.2)
Sufficient >75 nmol/L	10 (9.9)	6 (6.4)

Data are *n* (%) or median (25th, 75th percentile). mITT, modified intention-to-treat; BMI, Body Mass Index; MUAC, Mid-Upper-Arm-Circumference; WBC, white blood cell count; ESR, erythrocyte sedimentation rate; 25(OH)D_3_, 25-hydroxyvitamin D.

**Table 3 nutrients-11-00133-t003:** HIV viral load in placebo versus vitD_3_ + PBA.

			Crude			Adjusted ^1^		
Endpoint	Week	*n*	Difference	95% CI	*p*-Value	Difference	95% CI	*p*-Value
All patients (mITT)								
HIV viral load	4	197	−0.14	(−0.42 to 0.13)	0.298	−0.12	(−0.39 to 0.15)	0.385
	8	197	0.14	(−0.10 to 0.38)	0.254	0.15	(−0.09 to 0.39)	0.214
	16	197	0.16	(−0.10 to 0.42)	0.234	0.17	(−0.09 to 0.42)	0.205
	24	197	0.23	(−0.07 to 0.52)	0.134	0.28	(−0.01 to 0.56)	0.056
Patients (per-protocol)								
HIV viral load	4	180	−0.15	(−0.44 to 0.13)	0.288	−0.11	(−0.39 to 0.18)	0.459
	8	178	0.17	(−0.09 to 0.42)	0.201	0.18	(−0.07 to 0.44)	0.164
	16	173	0.19	(−0.08 to 0.46)	0.174	0.21	(−0.07 to 0.48)	0.137
	24	153	0.26	(−0.07 to 0.59)	0.115	0.35	(−0.03 to 0.67)	0.031

^1^ Data are adjusted for gender, age, and CD4 T cell counts and HIV viral load at baseline. CI, confidence interval; mITT, modified intention-to-treat.

**Table 4 nutrients-11-00133-t004:** Adverse events.

Manifestation (no)	Placebo (*n* = 39)	VitD_3_ + PBA (*n* = 35)
URTI	3	2
Acute bronchitis	1	1
Pneumonia	1	1
UTI	4	6
Otitis media	2	0
Tonsillitis	0	2
Lymphadenitis	1	1
Vaginal candidiasis	1	1
Herpes zoster	0	2
Skin infection	0	2
Dental caries	3	0
Oral rash	0	1
Carbuncle	3	3
Acute febrile illness	2	2
Sweating	0	1
Fatigue	1	1
Cough	6	3
Loss of appetite	2	1
Skin itching	0	1
Asthma	0	1
Arthralgia	3	2
Neuralgia	0	1
Anxiety disorder	1	0
Headache	0	2
Insomnia	0	1
Diarrhea	5	2
Constipation	1	0
Abdominal cramp	4	2
Dyspepsia	5	4
Numbness	0	1
Allergic conjunctivitis	1	0
Allergic dermatitis	1	0
Amenorrhea	1	0
Vaginal discharge	1	1
Total AEs	53	48

All AEs were grade 1 or mild, apart from herpes zoster (maculopapular rash) (2), constipation (1), diarrhea (1), oral rash (1), and insomnia (1), which were classified as grade 2 AEs. URTI, upper respiratory tract infection; UTI, urinary tract infection; AE, adverse event.

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
