# Peer review of "Daily Nutritional Supplementation with Vitamin D_3_ and Phenylbutyrate to Treatment-Naïve HIV Patients Tested in a Randomized Placebo-Controlled Trial"

_nutrients, 2019, doi:10.3390/nu11010133_

Reviewer 1 Report

In the present paper, Ashenafi et al. present the findings of a randomized, double blind, placebo controlled,  clinical trial on daily nutritional supplementation with vitamin D3 and phenylbutyrate to treatmentnaïve HIV patients.

The authors fail to see significant changes in HIV RNA load and CD4+ T-cell counts over a 16-week study period. 

 Introduction

Lines 51-52 contain strong statements; sentences should be rephrased given the wide range of clinical presentations of HIV-infected subjects

Lines 60-62: current American and European guidelines recommend cART introduction regardless the CD4+ T-cell count; if the authors refer to other guidelines this should be clearly stated and motivated in the text.

 Methods

Randomized, doubleblind, placebocontrolled, clinical trial with adequate sample size and statistical analysis. My major concerns are the following:

-       the patient-population enrolled (naïve subjects vs cART-treated) and the definition of slow progressors which is not used in clinical practice (references are not updated)

-       16 weeks of follow-up are not enough to detect significant differences in viral load and CD4 counts in antiretroviral naïve subjects. If the authors have stored plasma/cell samples of the patients enrolled in the study, inflammation and/or T-cell activation parameters may be measured, thus adding to the papers’ findings.

 - No data on clinical biomarkers (e.g. inflammation) are provided

Results

Clearly presented.

 Discussion

This section is too long and contains speculations which are not supported by the author’s findings: specifically, the authors do not provide any data on the possible mechanisms by which vit D3/PBA should decrease viral replication.

Line 285: current consensus is to start therapy as soon as possible and there is no rational to delay cART introduction.

Line 296: vit D3 supplementation is routinely prescribed in clinical practice 

Author Response

Reviewer 1:

 Comments:

 1.      In the Introduction, lines 51-52 contain strong statements; sentences should be rephrased given the wide range of clinical presentations of HIV-infected subjects.

Reply: We have omitted the sentence, line 51-52, and instead rewritten the first paragraph of the Introduction (underlined text, page 2, line 47-67) to describe the different phases of HIV infection and rates of disease progression including slowly progressive HIV infection.

 2.      Introduction, lines 60-62: current American and European guidelines recommend cART introduction regardless the CD4+ T-cell count; if the authors refer to other guidelines this should be clearly stated and motivated in the text.

Reply: We agree with the referee that the current guidelines for start of ART should be clarified in the Introduction. We have now changed the text in the Introduction (underlined, page 2, line 62-67): “WHO guidelines from 2015 recommend initiation of ART to HIV-infected individuals as soon as possible, regardless of CD4+ T cell counts, to reduce the morbidity and mortality associated with HIV infection. While the previous guidelines instructed to start ART at CD4+ T cell counts<350 c/ml, these recommendations were revised based on evidence from clinical trials showing thatearlier initiation of ART could delay a decline in CD4+ T cell counts and prevent immunological deterioration.” We have also added the WHO guidelines from Sept, 2015, as a new reference 8 in the reference list.

 To clarify the rationale and inclusion criteria for the current trial that was conducted before the revision of the WHO guidelines in Sept, 2015, we have also added a sentence at the end of the Introduction (underlined, page 2, line 88-89): “At the time this clinical trial was conducted, the national guidelines in Ethiopia was to initiate ART in HIV patients with clinical symptoms and a CD4+ T cell count <350 cells/µl. Therefore,ART-naïve HIV-positive individuals with slow progressive disease were enrolled…”

 3.      Concerns in methods are the patient-population enrolled (naïve subjects vs cART-treated) and the definition of slow progressors which is not used in clinical practice (references are not updated).

Reply: We would like to underline that this study did not enroll any HIV patients on ART, but this study focused solely on individuals who had not yet started ART. We did include a group of n=52 uninfected controls, who were not part of the randomized trial but were only sampled to receive relevant controls of baseline variables such as CD4+ and CD8+ T cell counts, BMI, vitD status etc.   

 Moreover, it is correct that it is important to be clear with the definition and how to use the definition of HIV slow progressors. Therefore we have omitted the last sentence in paragraph 3.2. Baseline characteristics, and instead included the following sentence in the Results (underlined, page 5, line 203-205): “These results confirmed that enrolled study participants had progressive HIV infection, but with viral loads in the lower range.” We have also omitted part of a sentence in the Discussion, page 12, line 359-360. Please, also see the changes we made with regards to describing HIV disease in the text of the Introduction at page 2.

However, the concept of slow progression (ie. long-term slow- or non-progressors) in HIV is not new (PMID: 26051387, 8905100, 17502001). In HIV research it could make sense to categorize patients this way, as has been done before (PMID: 23211771, 17484217, 15507395). In our study, we enrolled HIV-infected individuals who mostly managed to control HIV replication, despite the absence of ART, which indirectly suggested that these HIV patients had a more slowly progressive disease. This is the only conclusion about the study cohort that we make, based on the clinical characteristics and laboratory results that we have available. In a way, this study is completely unique, since these type of studies wouldn’t be feasible today when WHO guidelines recommend all HIV patients to start ART.

4.      16 weeks of follow-up are not enough to detect significant differences in viral load and CD4 counts in antiretroviral naïve subjects. If the authors have stored plasma/cell samples of the patients enrolled in the study, inflammation and/or T-cell activation parameters may be measured, thus adding to the papers’ findings. In addition, no data on clinical biomarkers (e.g. inflammation) are provided

Reply: We agree with the referee that it would have been beneficial to be able to supplement (4 month) and follow (6 months) these HIV patients for substantially longer period of times, at least for a year. But this is not the way the study was initially designed, when the hypothesis was that the effect of vitD and PBA would come rather early, to enhance innate mucosal defense mechanisms in the gut epithelia that would results in reduced immune activation, bacterial translocation and eventually restore CD4+ T cell counts and reduce HIV viral load. We collected a vast amount of clinical samples from this study cohort, including longitudinal PBMCs (frozen and in mRNA later), plasma and gut tissue biopsies, from HIV patients as well as the uninfected controls, to be able to study the immune response in the treatment group versus the placebo group in more detail. We have another manuscript in preparation, where we have analyzed sCD14 and LL-37 levels in plasma samples and performed microbiota analyses on part of the biopsy material. The results imply immune activation and changes consistent with HIV infection, such as a skewed microbiota in HIV infected versus uninfected controls, but no significant differences comparing treatment with placebo. Next, we plan to continue these analyses with cytokine/chemokine profiles in plasma and also study how multifunctional and regulatory T cell responses differ in these groups. However, this is extensive work that will be published separately. As customary for clinical trials, we´d like to report the clinical primary and secondary outcomes, adverse event and blood chemistry analyses as predefined in the clinical trial protocol. But we have added a sentence in the manuscript discussion (underlined, page 12, line 370-372): “Secondary analyses of inflammation and T cell activation could add additional information on the response to vitD3+PBA treatment in these patients.   “

 5.      The Discussion is too long and contains speculations which are not supported by the author’s findings: specifically, the authors do not provide any data on the possible mechanisms by which vitD3/PBA should decrease viral replication.

Reply: In principle, we have followed the recommended outline for clinical trial Discussions (PMID: 10231230), where it is suggested that there should be a discussion around possible mechanisms. But we agree that it is perhaps not so relevant to add discussions on mechanisms that was rather the basis for the study objectives than a mechanism being able to describe the study outcome. Therefore, we have decided to exclude (strikethrough lines in the Discussion) the most exploratory part of the Discussion involving possible effects of vitD and PBA on the antimicrobial host response.

6.      Line 285: current consensus is to start therapy as soon as possible and there is no rational to delay cART introduction.

Reply: Yes, we agree and this part of the sentence has been omitted. Please, note that this study was designed and conducted before the new WHO guidelines were implemented.

 7.      Line 296: vitD3 supplementation is routinely prescribed in clinical practice.

Reply:  As far as we know, there is no prescription for vitD in low-income countries such as Ethiopia. Furthermore, in countries where it is prescribed, the doses are generally substantially lower than 5000 IU/day.

Reviewer 2 Report

General comments

 In this report the authors have tested daily vitD3 +PBA nutritional supplementation to reduce viral replication and restore immune and nutritional status in anti-retrovirus treatment naïve HIV infected patients. This study is well designed and the article well written. The intervention is supported by adequate background information and prior data. The use of the modified intention to treat statistics is legitimate in the context of the study.

 Specific comments

 Being issued from the same population, somewhat surprisingly HIV+ participants exhibit a higher median serum 25OHD concentration than non-infected individuals. Were sampling done during the same period of the year? Could the difference in sex ratio be a factor? Can it be accounted for? Also the definition of vitamin D deficiency with the 50 nmol/L threshold is disputable. Some guidelines suggest 30 nmol/L. It would be interesting and useful, if possible, to compare the response of participants truly deficient (≤30 nmol/L) to those replenished (≥75) at baseline in terms of the immunology tests & HIV load. Or could data be adjusted for serum 25OHD at entry in addition to gender, age, CD4 T cell counts? Would this add pertinent information?

 Author Response

Reviewer 2:

 Comments:

 1.      Being issued from the same population, somewhat surprisingly HIV+ participants exhibit a higher median serum 25OHD concentration than non-infected individuals. Were sampling done during the same period of the year? Could the difference in sex ratio be a factor? Can it be accounted for?

Reply: This is a correct and interesting observation made by the referee. Sampling was actually performed in parallel with the sampling of HIV-positive study subjects, meaning the same time of year. We have recently shown in another study including patients with tuberculosis (TB) and/or HIV as well as uninfected controls, that regardless of infection, vitD status is significantly lower in Ethiopian individuals at a lower latitude compared with Swedish individuals at a higher latitude (PMID: 29867045). Basically, this phenomenon could be explained by many different factors, but most importantly, vitD deficiency or insufficiency is not a problem only among HIV patients, but it is a problem in the general population. As we described in a recent review (PMID: 29804293), the vitD status can vary substantially between different countries, and the exact reasons for this is yet to be determined. However, high-dose vitD supplementation can only be justified in populations with vitD deficiency or insufficiency, and it has also been shown that vitD is most beneficial in patients with low vitD status (PMID: 28202713 and 29696707).

        The importance of gender in clinical vitD trials is rather unexplored, but may affect baseline vitD status as well as the response to vitD supplementation. In this study, we were surprised to find a significant over-representation of females, as also discussed in the manuscript. It is correct that we had more males than females in the control group, resulting in a significant difference in gender distribution (p<0.001) and the control group also had significantly lower vitD status (p=0.003). Low level of vitD in females may be associated to pregnancy or menopause, while low vitD in males could be induced by eg. prolonged alcohol consumption. Interestingly, a recent study comparing vitD status in males and females enrolled in a large Indian cohort (3,879 participants), found that vitD status indeed was significantly lower in males (PMID: 28315864). The proportion of males with vitD levels<30 nmol/l, was also significantly higher among males compared to females; 58% compared to 51%, respectively. This reference and new text (underlined) has now been added to the manuscript discussion, page 11-12, line 338-344: “In addition, we found that the uninfected control group contained more males than females, and the controls had significantly lower vitD3 status compared with the HIV-infected subjects. Consistently, a recent study comparing vitD3 status in males and females, enrolled in a large Indian cohort (3,879 participants), found that vitD3 status was significantly lower in males who also had a higher incidence of vitD3 deficiency (<30 nmol/l) (PMID: 28315864). The importance of gender in clinical vitD3 trials is rather unexplored, but may affect baseline vitD3 status as well as the response to vitD3 supplementation and should therefore be further explored.”

          In our manuscript, all statistical analyses were based on crude and adjusted data sets, and gender was one of the covariates adjusted for (together with the other variables which we a priori believed to be correlated with the outcome). The importance of gender and vitD status will, however, be an interesting research question for future vitD trials. 

 2.      Also, the definition of vitamin D deficiency with the 50 nmol/L threshold is disputable. Some guidelines suggest 30 nmol/L. It would be interesting and useful, if possible, to compare the response of participants truly deficient (≤30 nmol/L) to those replenished (≥75) at baseline in terms of the immunology tests & HIV load. Or could data be adjusted for serum 25OHD at entry in addition to gender, age, CD4 T cell counts? Would this add pertinent information?

Reply: We understand that the thresholds set for vitamin D deficiency may not be straight forward. Basically, vitD thresholds used in bone health and to prevent infection may be different. It is also a matter of debate between US Institute of Medicine and the Endocrine Society (PMID: 29804293), but for the treatment of infection we have chosen to follow the guidelines provided by the Endocrine Society (PMID: 21646368 and also 28516265), defining vitD deficiency as<50 nmol/l. The Endocrine Society reference is provided in the manuscript, page 6, line 219. Nevertheless, we agree that vitD levels <30 nmol/l could be regarded as severe vitD deficiency. We have performed these analyses and other sub-group analyses (result section, page 7, line 234-236) with our data sets, but it does not change the overall statistical significance. Perhaps a larger samples size could justify these sub-group analyses, but currently we don´t believe that would change the results. Nor did adjustment for vitD levels at baseline reveal any statistical differences; perhaps because the vitD levels are very similar between treatment and placebo.

 Round  2

Reviewer 1 Report

The authors have addressed all my concerns.